# X-ray Single Exposure Imaging and Image Processing of Objects with High Absorption Ratio

**DOI:** 10.3390/s23052498

**Published:** 2023-02-23

**Authors:** Yanxiu Liu, Kaitai Li, Dan Ding, Ye Li, Peng Zhao

**Affiliations:** 1College of Physics, Changchun University of Science and Technology, Changchun 130022, China; 2Electronic Information Engineering College, Changchun University, Changchun 130022, China

**Keywords:** high absorption ratio, single-exposure, X-ray imaging, image contrast enhancement

## Abstract

The dynamic range of an X-ray digital imaging system is very important when detecting objects with a high absorption ratio. In this paper, a ray source filter is used to filter the low-energy ray components which have no penetrating power to the high absorptivity object to reduce the X-ray integral intensity. This enables the effective imaging of the high absorptivity objects and avoids the image saturation of low absorptivity objects, thus achieving single exposure imaging of high absorption ratio objects. However, this method will reduce the image contrast and weaken the image structure information. Therefore, this paper proposes a contrast enhancement method for X-ray images based on Retinex. Firstly, based on Retinex theory, the multi-scale residual decomposition network decomposes the image into an illumination component and a reflection component. Then, the contrast of the illumination component is enhanced through the U-Net model with the global–local attention mechanism, and the reflection component is enhanced in detail using the anisotropic diffused residual dense network. Finally, the enhanced illumination component and the reflected component are fused. The results show that the proposed method can effectively enhance the contrast in X-ray single exposure images of the high absorption ratio objects, and can fully display the structure information of images on devices with low dynamic range.

## 1. Introduction

In the process of digital radiography (DR), objects with a high absorption ratio need higher X-ray photon energy and exposure, while objects with a low absorption ratio need the opposite [1]. This requires an X-ray flat panel detector (FPD) to have a high dynamic range (HDR). The dynamic range of the FPD determines the imaging effect of objects with a high absorption ratio. In fact, the dynamic range of the FPD is limited [2]. When high absorptivity objects can be effectively imaged and resolved, images of low absorptivity objects usually reach saturation and cannot be resolved. Therefore, it is necessary to use an X-ray HDR imaging method to solve the imaging problem of objects with a high absorption ratio [3].

At present, there are two main methods for X-ray HDR imaging [4]. One is that the low-dynamic-range imaging device is used to make multiple exposures of the same scene with variable energy, and then the obtained multiple images are fused into an HDR image [5]. Another method is that the optical path expansion technology is used to expand the light intensity information in the same scene into multiple optical signals, the exposure parameters of imaging devices in different optical paths are set to achieve imaging, and the output images of multiple sensors are fused into an HDR image with complete data information [6].

X-ray HDR imaging of high absorption ratio objects usually uses multiple exposures of different energies. Image fusion keeps the X-ray image information of high absorptivity and low absorptivity objects in one image [7]. In recent years, Chen et al. fused sequences of different voltages at each projection angle, thus extending the dynamic range of the X-ray imaging system [8]. Haidekker et al. proposed a mathematical framework for recovering absorbance from images of objects with high absorption ratios, thereby enhancing the dynamic range of X-ray imaging [9]. Li et al. combined self-adaptive multi-exposure imaging technology and HDR image fusion, and proposed a method for extending the dynamic range of X-ray imaging systems based on linear constraints with variable energy [10].

The above research shows that multi-exposure imaging technology is an effective method to expand the dynamic range of the X-ray imaging system. However, the exposure time and image transmission time of DR are relatively long, resulting in the low detection efficiency of multi-exposure, and the change in the geometric position of the object to be measured will lead to image fusion difficulties [11]. At the same time, the complex imaging system structure of optical path extension technology makes it difficult to determine the exposure conditions [12]. Based on the above reasons, this paper proposes a single exposure imaging technique for objects with a high absorption ratio based on X-ray spectral bandwidth compression. In order to achieve the compression of the X-ray spectral bandwidth, a ray source filter is used to filter the low energy rays that cannot pass through high absorptivity objects, thus reducing the integral intensity of the X-rays that pass through low absorptivity objects and avoiding image saturation. This is equivalent to compressing the dynamic range of the image at the X-ray image end, which will reduce the image contrast. Aiming for the features of such images, in this paper, an X-ray image contrast enhancement method based on Retinex is designed, which realizes that all information can be obtained via the single imaging of objects with a high absorption ratio.

To be specific, a multi-scale residual network is designed to decompose an image into two components (the illumination component and the reflection component). A global–local attention mechanism is proposed for U-Net networks to enhance the contrast in illumination components. An anisotropic diffusion-based residual dense network is designed to deal with the reflection component. Meanwhile, reconstruction loss, illumination smoothness loss, and constant reflectivity loss are introduced to train the decomposition network. Structural similarity loss is introduced to train the enhanced network.

The main ideas and contributions of this paper are summarized as follows:X-ray spectrum bandwidth compression is realized by using a ray source filter, which avoids the image saturation of low absorptivity objects.In view of the low contrast of X-ray images, this paper proposes an X-ray contrast enhancement method based on Retinex. According to the invariant property of the reflection component, the decomposition network is trained using single exposure and multiple exposure X-ray images.Aiming at the phenomenon of uneven global brightness in the illumination components, this paper proposes a U-Net network based on the global–local attention mechanism to extract the illumination features of different scales, and use the complementarity of global and local features to enhance the contrast in the illumination components.In order to avoid ghost artifacts and detail blurring of reflection components, this paper designs a residual dense network based on anisotropic diffusion, which uses hierarchical information to enhance detailed information while suppressing noise and artifacts.

The rest of this paper is organized as follows: In Section 2, the influence of the filter on the X-ray spectrum is introduced and the experiment is carried out. In Section 3, the network structure of the proposed contrast enhancement algorithm is analyzed, and the effectiveness of the method is verified via simulation experiments. Finally, the conclusion is given in Section 4.

## 2. X-ray Spectral Bandwidth Compression and Imaging

X-ray imaging equipment is composed of an X-ray tube, high voltage power supply, and X-ray FPD. In this paper, the tungsten target X-ray tube (tube voltage 180 kV, tube current 1 mA, focus size 0.5 mm) was used, the FPD size was 17 inches × 17 inches, the resolution was 3072 × 3072, and the scintillator was CsI crystal. The exposure parameters were tube current of 0.5 mA, exposure time of 2 s, and adjustable tube voltage. The geometry and position of the X-ray tube, imaging sample, and device are shown in Figure 1.

### 2.1. Effect of Filter on X-ray Spectrum

The absorption of X-rays by substances conforms to the Lambert–Beer law, that is, the intensity of emitted X-rays decays exponentially with the thickness, and the intensity of the X-rays passing through the object is:(1)$$I=I_{0}e^{-\mu_{m}\cdot \rho x}$$
where $I_{0}$ is the initial X-ray intensity, $\mu_{m}$ is the mass absorption coefficient, and x is the thickness of the object. In the traditional model, the mass absorption coefficient can be expressed as:(2)$$\mu_{m}=K\lambda^{3}Z^{4}$$

In the formula, K is the constant, *λ* is the X-ray wavelength, and Z is the atomic number. As can be seen from the formula, for a substance with atomic number Z, the mass absorption coefficient rapidly decreases exponentially with the decrease in X-ray wavelength. The nonlinear attenuation of low energy rays and high energy rays can be realized by utilizing the nonlinearity of X-ray absorption by matter, and the purpose of X-ray spectral bandwidth compression can be achieved. Figure 2 shows the relationship curve of the X-ray radiation spectrum with the filter plate at the maximum energy of 180 kV (to simplify the calculation, the characteristic emission peak of the target material and the characteristic absorption peak of the material are ignored). It can be seen from the comparison of the X-ray spectrum curves transmitted by three filters that the larger the atomic number of the filter material, the narrower the X-ray spectrum width of the filter.

By using the filter to realize the X-ray spectral width compression characteristic, the low energy X-ray can be greatly attenuated, while the high energy X-ray can be mostly retained, thus greatly reducing the X-ray integral intensity of penetrating low absorptivity objects and avoiding the saturation of the FPD.

### 2.2. Analysis of the Influence of Narrowband X-ray on Imaging

#### 2.2.1. Image Gray Distribution Test

The step test block shown in Figure 3a was used for the imaging test. The filter material is silver, and the dynamic range of the FPD image is 16 bits. After imaging, the gray distribution curve of the iron test block and aluminum test block is obtained according to the position shown in Figure 3b.

Figure 4a,b shows the gray distribution curve of iron and aluminum test blocks under 90 kV exposure. The effective penetration depth of the iron test block is 8 mm, and the other six steps cannot be effectively imaged. The aluminum test block can be effectively imaged under various conditions. Figure 4c,d shows the gray distribution curve of the iron test block and aluminum test block under a 180 kV exposure condition. The iron test block can be effectively imaged under various conditions. Under the condition of no filter, the steps of the aluminum test block are saturated and cannot be effectively imaged, that is, the dynamic range of the X-ray transmission intensity exceeds the effective dynamic range of the FPD. With the increase in the thickness of the filter, the saturation phenomenon of the aluminum test block is gradually weakened. When the thickness of the filter reaches 1 mm, the thinnest part of the aluminum test block can also be effectively imaged, achieving a single exposure imaging of 20 mm iron and 1 mm aluminum. The results show that the X-ray image with an HDR can be well compressed into the effective dynamic range of the FPD by using the filter, so as to realize the single exposure imaging of objects with a high absorption ratio.

#### 2.2.2. Luminance Compression Ratio

The ratio of the gray value of the image without the filter to the gray value with the filter is defined as the luminance compression ratio. The thickness of the test block that can be imaged normally under different exposure conditions is counted, and the results are shown in Figure 5. It can be seen from the curve that the luminance compression ratio increases with the increase in the thickness of the filter, and the smaller the absorbance is, the greater the compression ratio is. This indicates that the proportion of the remaining high energy rays in the filtered X-ray spectrum increases, the proportion of the integral intensity of the rays transmitted by high absorptivity objects increases, and the proportion of the integral intensity of the X-ray transmitted by low absorptivity objects decreases.

#### 2.2.3. Image Contrast Feature

The use of a filter can achieve single exposure imaging of objects with a high absorption ratio, but with the compression of the dynamic range of X-ray images, the image contrast will be reduced. Under different exposure conditions, there are impenetrable phenomena of iron test blocks and image saturation phenomena of aluminum test blocks. Therefore, it is impossible to calculate the contrast generated by a gray scale of 20 mm iron and 1 mm aluminum. To this end, steps that can be distinguished under different exposure conditions are used for contrast statistics. In this paper, 4 mm iron and 6 mm iron are used. As shown in Figure 6, the image contrast decreases with the increase in the filter thickness.

In order to verify the change rule of the global maximum contrast of the image, take the gray value of the image at the thickest part of iron and the gray value of the image at the thinnest part of aluminum for the calculation. Under 150 kV and 180 kV exposure conditions, 20 mm iron can effectively be imaged, so only these two sets of data are taken. The “---” symbol in the table represents image saturation and cannot be calculated, so only partial data can be obtained in global contrast statistics. The rule in Figure 6 can also be obtained from Table 1, that is, under the same exposure voltage, the contrast decreases with the increase in filter thickness and decreases with the increase in working voltage. Contrast is one of the important parameters of the image, so it is necessary to enhance it to improve the human eye recognition ability.

## 3. X-ray Image Contrast Enhancement Method Based on Retinex

Inspired by the Retinex theory [13], a deep Retinex network was designed, and the overall architecture is presented in Figure 7. The network consists of three subnetworks: illumination and reflection decomposition network, illumination adjustment network, and reflection enhancement network.

The original image S was first decomposed into illuminance $\tilde{I}$ and reflectance $\tilde{R}$:(3)$$\left[\tilde{I},\tilde{R}\right]=f_{decom}\left(S\right)$$
where $f_{decom}\left(\;\right)$ signifies the image decomposition process. Illumination I represents the various lightness on objects. Reflectance R describes the intrinsic property of captured objects, which is considered to be consistent under any lightness conditions.

Then, the decomposed $\tilde{I}$ and $\tilde{R}$ were enhanced to obtain the enhanced illumination component $\hat{I}$ and the restored reflection component $\hat{R}$:(4)$$\hat{I}=f_{enhance}\left(\tilde{I}\right)$$
(5)$$\hat{R}=f_{restore}\left(\tilde{R}\right)$$

Finally, the reconstructed image $\hat{S}$ was obtained via element-wise multiplication:(6)$$\hat{S}=\hat{I}\cdot \hat{R}$$

### 3.1. Image Decomposition Network

At the decomposition stage, a multi-scale residual decomposition network (Decom-Net) was designed to generate the illumination component and the reflection component, as shown in Figure 8. Decom-Net uses a 3 × 3 convolution layer to generate the characteristic of the input image. Then, it uses six convolution layers with the activation function rectified linear unit (ReLU) to change the size of the feature map and learn the characteristics of the illumination part and the reflection part. After the second convolution, multi-scale feature extraction was added. Finally, convolution and Sigmoid function were used to map the learned image features onto illuminated images and reflected images, and then output them.

In the training process, paired single/multiple exposure images were taken as the inputs of Decom-Net. Since single and multiple exposure images of the same scene have the same reflectance, when they are fed into two Decom-Nets, the network weights are shared between them. In the testing process, only the single exposure image needs to be inputted.

### 3.2. Illumination Enhancement Network

The constructed illumination enhancement network (Enhance-Net) adopts the U-Net structure, which is composed of an encoder and decoder, as shown in Figure 9. In the encoder part, the input image is first encoded into a lower dimension and then entered into the global–local attention mechanism. Finally, the original resolution image is obtained via input to the decoder. The encoder–decoder structure is composed of four convolution blocks. The difference is that the encoder is composed of two convolution layers and a pooled down-sampling layer, and the decoder is composed of two convolution layers and a deconvolution up-sampling layer.

The global and local relationships in the illumination component have an important impact on the training of the network. In order to enhance the modeling ability of the network, the global–local attention mechanism (GLAM) is proposed, and its structure is shown in Figure 10. Enhance-Net integrates GLAM to filter the features inputted into the network, enhancing useful information and suppressing invalid information.

Global attention mechanism: Because the illumination distribution is not uniform and consistent, the designed global attention mechanism builds the relationship between each channel, making the illumination component uniform on the overall pixel. By calculating the covariance between channels in the feature map, the global correlation and the covariance matrix are obtained. Then, the matrix is row convolved and normalized to retain the structure information, and the attention weight of each channel is obtained. Finally, the original feature map is multiplied by the attention weight.

Local attention mechanism: Since different objects have different features in the illumination component, the local information of the illumination component is used to enhance contrast. Firstly, the feature graph is divided into several feature blocks of the same size, and the mean and standard deviation of the feature blocks are used to improve the ability of capturing feature information. The mean value reflects the brightness of the image. The standard deviation reflects the dispersion degree between the image pixels and the mean value, and the clarity degree of the edge of the illumination component. The features after the mean and standard deviation are convolved and normalized to obtain the weight map of the feature block, and then multiplied with the original feature block to obtain local features.

### 3.3. Reflectance Restore Network

As for the reflectance, it is usually contaminated by the blurring of details; the reflectance restoration network (Restore-Net) is designed to remove noise and enhance details, as shown in Figure 11.

In Restore-Net, reflectance passes through three residual dense blocks, each focused on a task without interruption by other groups. The central convolution layer in the residual dense block is set into two groups, which avoids influence from the previous layer and makes effective use of the hierarchical information. Image edge information is enhanced by anisotropic diffusion filtering (AD). Dense connections can effectively boost performance and preserve semantic information. To avoid the disappearance of network gradients, a residual skip connection to connect all of these dense blocks as well as the successive concatenation and convolution layer is added.

### 3.4. Loss Function

#### 3.4.1. Decomposition Process

Motivated by Retinex-Net [14], the loss functions $L_{Decom}$ of Decom-Net include reconstruction loss $L_{recon}$, illumination smoothness loss $L_{is}$, and invariable reflectance loss $L_{ir}$.
(7)$$L_{Decom}=L_{recon}+\lambda_{is}L_{is}+\lambda_{ir}L_{ir}$$
where $\lambda_{is}$ and $\lambda_{ir}$ denote the coefficients of illumination smoothness and balance reflectance consistency, respectively.

Reconstruction loss: In Decom-Net, the similarity between the reconstructed results of the illumination and reflection components of the image’s own decomposition and the original image not only need attention. It is also necessary to pay attention to the similarity between the combined reconstruction results of the illuminance component and the reflection component of the paired image decomposition and the single/multiple exposure image. Reconstruction loss is shown in Equation (8):(8)$$L_{recon}=\underset{i=se,me}{\sum }\underset{j=se,me}{\sum }\lambda_{ij}\|R_{i}\circ I_{j}-S_{j}{\|}_{1}$$
where S is the original image, R is the reflection component, I is the illumination component, and $\lambda_{ij}$ is the weighting factor.

Illumination smoothness loss: The single/multiple exposure image decomposition does not have the same illumination components but should highlight the structure of the image and local details while maintaining overall smoothness. The traditional total variation (TV) minimization as a loss function will cause over-smoothing.
(9)$$L_{is}=\underset{i=se,me}{\sum }\|\nabla I_{i}\circ \exp\left(-\lambda_{g}\nabla R_{i}\right)\|$$
where ∇ represents the gradient (including horizontal and vertical gradient), $\lambda_{g}$ represents the gradient-aware balance coefficient, and $\exp\left(-\lambda_{g}\nabla R_{i}\right)$ relaxes the smoothing constraint at locations with more complex image structures and illumination discontinuities.

Invariable reflectance loss: According to the Retinex theory, it is known that the reflection components are invariant by the nature of the object itself, so the reflection components of single and multiple exposure images are similar. Furthermore, the invariable reflectance loss is used as the constraint of the training stage.
(10)$$L_{ir}=\|R_{se}-R_{me}{\|}_{1}$$
where $R_{se}$ is the reflection component of the single exposure image, $R_{me}$ is the reflection component of the multiple exposure image, and $\|{\|}_{1}$indicates the L1 norm operation.

#### 3.4.2. Illumination Enhancement Process

The illumination enhancement loss $L_{IE}$ maximizes the structural similarity between the illumination components before enhancement I and after enhancement $\hat{I}$ SSIM:(11)$$L_{IE}=-SSIM\left(\hat{I},I\right)$$

#### 3.4.3. Reflectance Restore Process

Reflectance component enhancement loss $L_{RE}$ constrains the enhanced reflection component $\hat{R}$. There is similarity between this and the initial reflection component R:(12)$$L_{RE}=-SSIM\left(\hat{R},R\right)$$

#### 3.4.4. Fusion Process

Fusion loss $L_{Fusion}$ limits the fusion image $\hat{I}\hat{R}$. There is similarity with the original image S:(13)$$L_{Fusion}=-SSIM\left(\hat{I}\hat{R},S\right)$$

### 3.5. Experiment and Analysis

In order to verify the effectiveness of the X-ray image contrast enhancement algorithm proposed in this paper, the proposed method was compared with other contrast enhancement algorithms. These include X-ray contrast enhancement methods, CLAHE [15], LCM-CLAHE [16], methods of contrast enhancement based on deep learning, Retinex-Net [14] and Zero-DCE [17].

#### 3.5.1. X-ray Images of Materials with High Absorption Ratio under Different Voltages

X-ray images of cables under different voltages (90 kV, 120 kV, 150 kV, and 180 kV) were selected. It can be seen from Figure 12 that the overall brightness of the original image is low, the contrast is not obvious, and the cable details are not prominent enough. With the increase in voltage, the internal structure of the cable gradually appears, but the overall brightness and contrast are still very low.


**Qualitative analysis**


Five methods were used to enhance the contrast in the original image. CLAHE and LCM-CLAHE improved the contrast of the image, but the brightness did not improve significantly. In the Retinex-Net and Zero-DCE methods, the image brightness is enhanced, but the overall over-exposure is not suitable for practical applications. The method proposed in this paper can reasonably improve the brightness and contrast of the image. At the same time, the detailed structure of the cable is also clearly visible, which is suitable for X-ray images with different voltages.


**Quantitative analysis**


The purpose of X-ray image enhancement is to enhance brightness and contrast and highlight details. Therefore, five evaluation indicators were selected to evaluate the performance of different methods. These included structure definition (NRSS), information entropy (entropy), spatial frequency (SF), average gradient (AG), and image standard deviation (STD). The higher their values, the better the image enhancement effect.

NRSS: reflects the clarity of the image;AG: reflects the detail change rate of the image, which can be used to characterize the clarity of the image;SF: reflects the overall spatial activity of the image;STD: reflects the dispersion between the gray value and the average value of each pixel in the image, which can be used to evaluate the contrast of the image;Entropy: represents the average information of the image, which is used to measure the information richness of the image.

Table 2 lists the quality indicators of each method for processing the above images, and the best value of these methods is highlighted in bold. It can be seen that the proposed method has the best evaluation index for all voltage values. Compared with the other four methods, the proposed method has higher clarity, contrast, and information richness.

#### 3.5.2. Qualitative Analysis of X-ray Images of Objects with Different Absorption Ratios


**Qualitative analysis**


Figure 13 and Figure 14 are X-ray images of the brake pump and water pump at 90 kV. It can be seen that the structures of the brake pump and water pump are complex. After CLAHE and LCM-CLAHE processing, the image does not improve significantly, and local details are not prominent enough. The images processed using Retinex-Net and Zero-DCE have clear details, but there are artifacts and the overall contrast is not enhanced. The proposed method improves the contrast and brightness of the image, and enhances the detail structure to a certain extent. The method is suitable for X-ray images with complex details.

Figure 15 and Figure 16 are X-ray images of an iron pipe at 120 kV and pressure doubling boxes at 100 kV, respectively. Among them, the iron pipe and pressure doubling box are defective, but the original X-ray image cannot be well displayed. After CLAHE and LCM-CLAHE, the defect is still dark and cannot be clearly displayed. Retinex-Net and Zero-DCE will blur the image, and the defect still has low contrast. In terms of the proposed method, in addition to the increase in global brightness, the contrast in defects is better than the other methods. Therefore, the proposed method can effectively reproduce the structure and defects in the dark area.


**Quantitative analysis**


Table 3 objectively reflects that the proposed method is very effective in improving the overall visual effect and the detail effect of the image. NRSS, AG, and SF reflect the sharpness of the image. Table 3 shows that the proposed method is clearer than the original image and other methods. STD reflects the contrast of the image, and the proposed method is far superior to other algorithms in contrast. Entropy reflects the richness in image information. The proposed method is superior to the other four methods in improving information entropy, which indicates that the method can retain more information. In conclusion, compared with the other methods, the proposed method achieved the best results in general.

## 4. Conclusions

The ray source filter is used to filter the low energy X-ray, realizing the compression of the X-ray radiation spectrum bandwidth, effectively reducing the ray integral intensity, penetrating the low absorptivity object, avoiding the saturation of the low absorptivity object image, and preserving all of the information of the high absorption ratio object in the single exposure image. In order to solve the problems of low contrast and structural information weakening caused by the dynamic compression of X-ray images, this paper proposes a Retinex global–local contrast enhancement model. In the decomposition phase, Decom-Net is constructed to decompose the original X-ray image into an illumination component and reflection components. In Enhance-Net, the global–local attention mechanism is used to adjust the contrast in the illumination components. In Restore-Net, anisotropic diffusion filtering is added to the residual dense network to enhance the image details and suppress noise and artifacts. Finally, the enhanced X-ray image is obtained by multiplying the enhanced illumination component and reflection component. The experimental results show that this method can effectively improve the visual effect and contrast, enhance detail, suppress noise, and avoid artifacts. The single exposure imaging method of objects with high absorption ratios used in this paper has higher detection efficiency and environmental adaptability than the traditional multiple exposure image fusion method, and has important practical value in portable X-ray detection equipment.

## Figures and Tables

**Figure 1 sensors-23-02498-f001:**
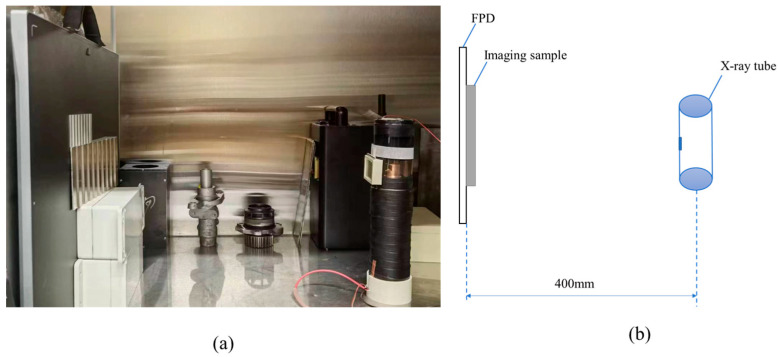
(**a**) X-ray imaging equipment, and (**b**) geometric location diagram of the equipment.

**Figure 2 sensors-23-02498-f002:**
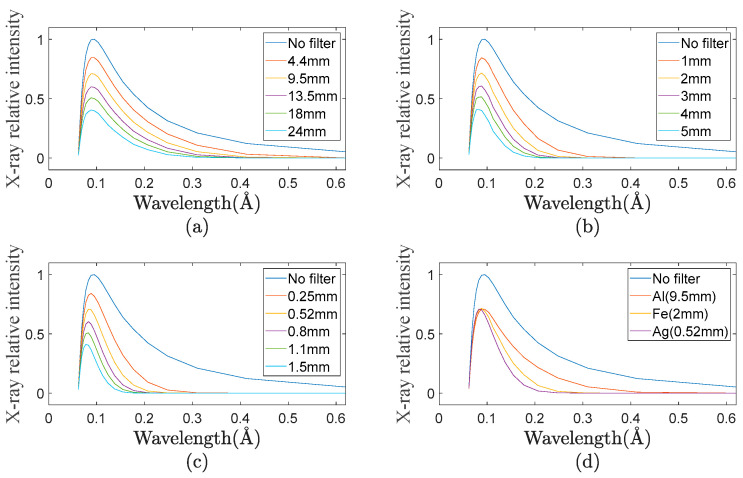
Influence curve of filter on X-ray spectrum. (**a**) Aluminum filter; (**b**) iron filter; (**c**) silver filter; and (**d**) comparison of filtering characteristics of three filters.

**Figure 3 sensors-23-02498-f003:**
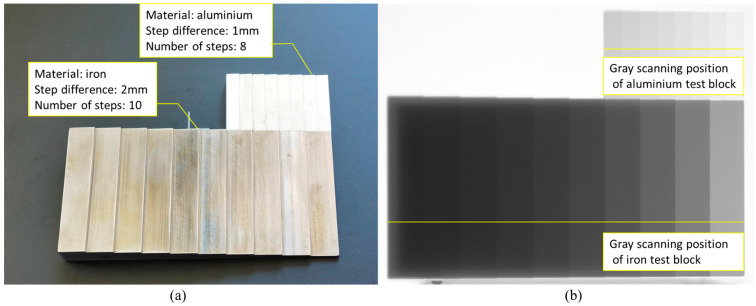
(**a**) Step test block used in imaging experiment and (**b**) obtaining position of gray curve of X-ray image of test block.

**Figure 4 sensors-23-02498-f004:**
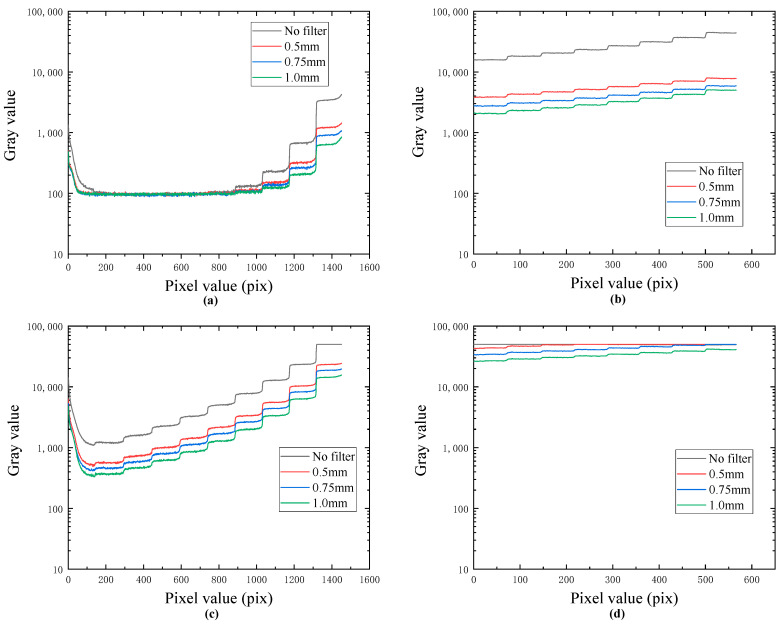
(**a**) Gray scale curve of iron image under 90 kV exposure; (**b**) gray scale curve of aluminum image under 90 kV exposure; (**c**) gray distribution curve of iron image under 180 kV exposure; and (**d**) gray distribution curve of aluminum image under 180 kV exposure.

**Figure 5 sensors-23-02498-f005:**
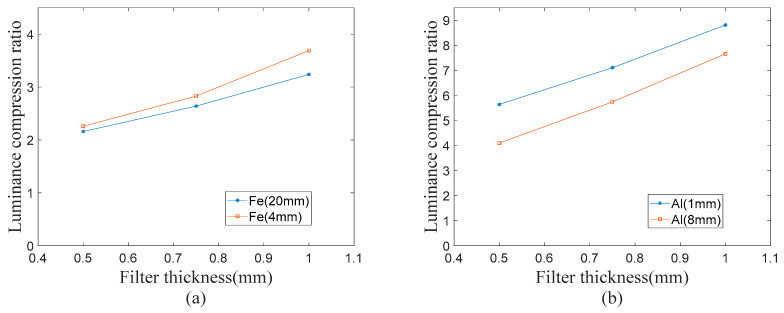
Relation curve between image brightness compression ratio and filter thickness: (**a**) 180 kV iron brightness compression ratio change curve and (**b**) 90 kV aluminum brightness compression ratio curve.

**Figure 6 sensors-23-02498-f006:**
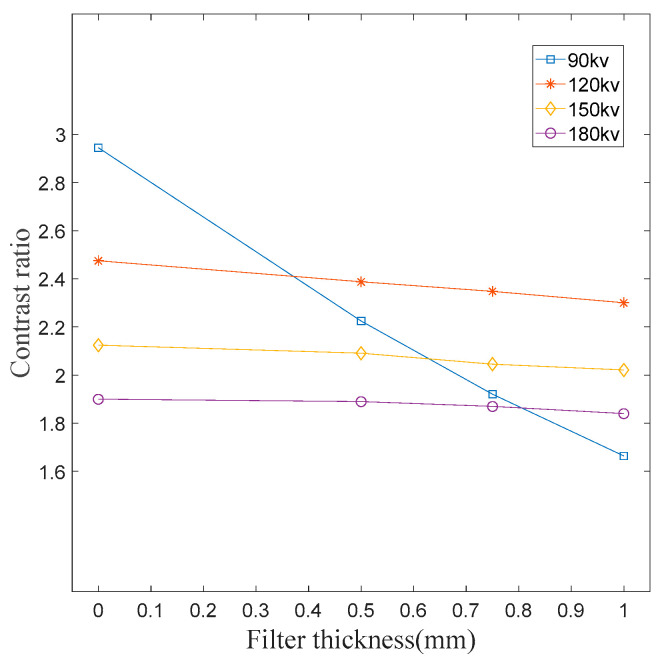
Relation curve between image contrast and filter thickness.

**Figure 7 sensors-23-02498-f007:**
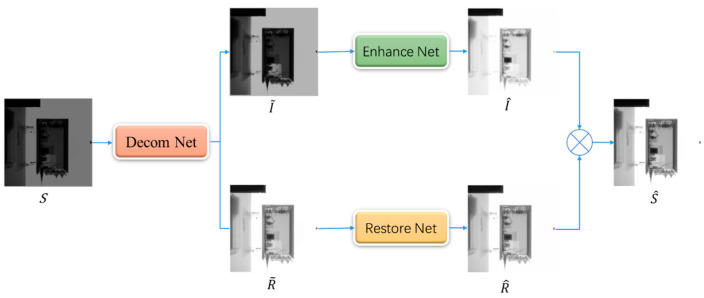
The network architecture of the proposed method.

**Figure 8 sensors-23-02498-f008:**
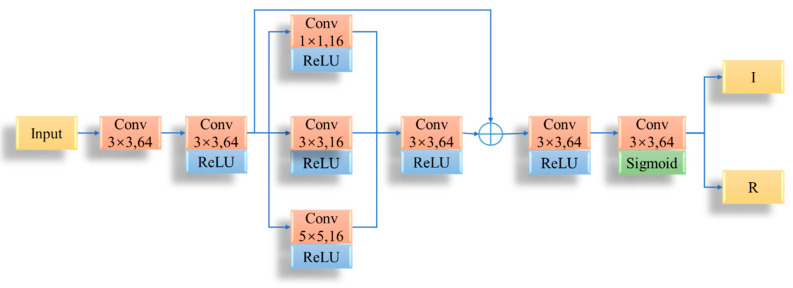
The structure of Decom-Net.

**Figure 9 sensors-23-02498-f009:**
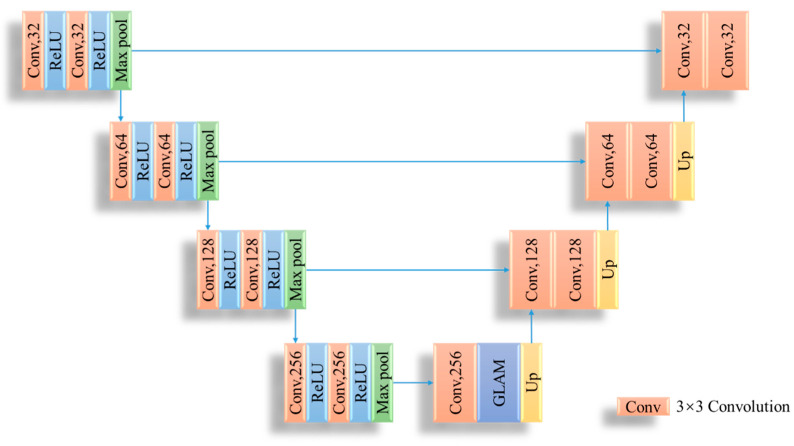
The structure of Enhance-Net.

**Figure 10 sensors-23-02498-f010:**
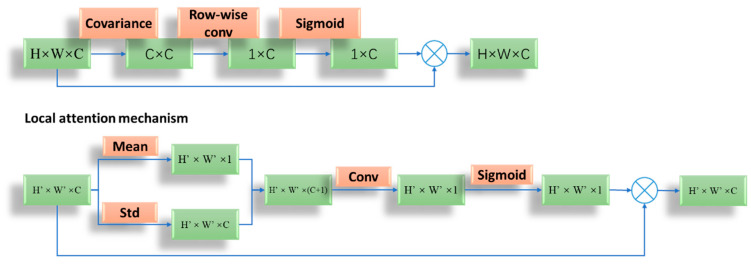
The detail architecture of GLAM.

**Figure 11 sensors-23-02498-f011:**
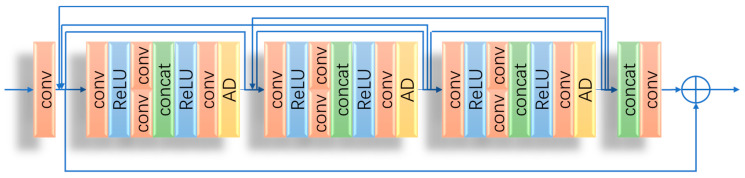
The structure of Restore-Net.

**Figure 12 sensors-23-02498-f012:**
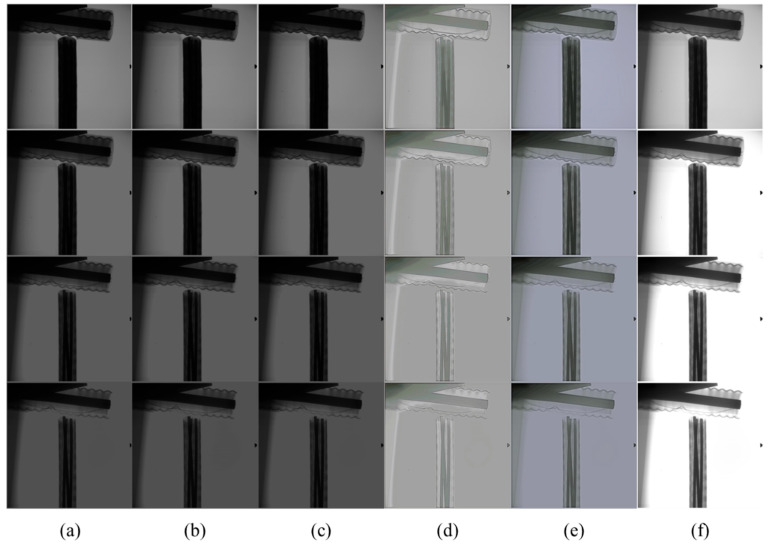
The contrast enhancement results of different methods (the voltage is 90 kV, 120 kV, 150 kV, and 180 kV from top to bottom). (**a**) The raw images; (**b**) CLAHE; (**c**) LCM-CLAHE; (**d**) Retinex-Net; (**e**) Zero-DCE; and (**f**) our method.

**Figure 13 sensors-23-02498-f013:**
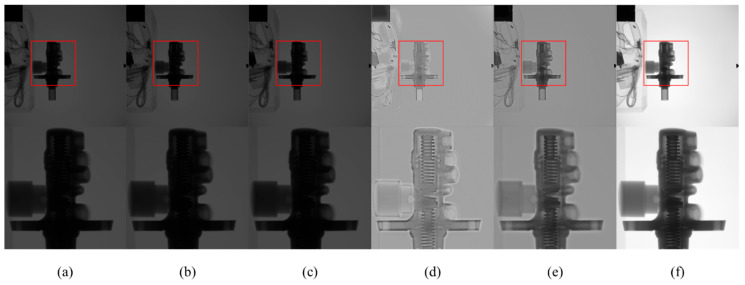
The contrast enhancement results of different methods. Top: raw image and the enhancement results; bottom: zoomed-in views on the highlighted area. (**a**) The raw images; (**b**) CLAHE; (**c**) LCM-CLAHE; (**d**) Retinex-Net; (**e**) Zero-DCE; and (**f**) our method.

**Figure 14 sensors-23-02498-f014:**
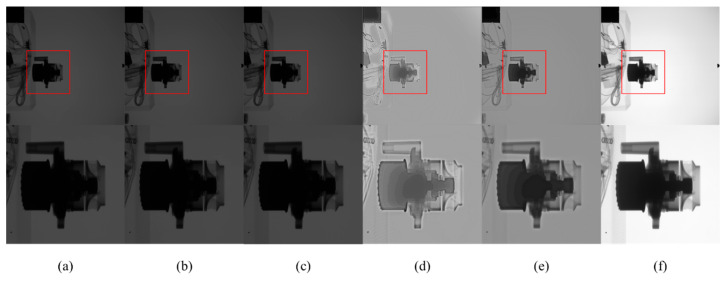
The contrast enhancement results of different methods. Top: raw image and the enhancement results; bottom: zoomed-in views on the highlighted area. (**a**) The raw images; (**b**) CLAHE; (**c**) LCM-CLAHE; (**d**) Retinex-Net; (**e**) Zero-DCE; and (**f**) our method.

**Figure 15 sensors-23-02498-f015:**
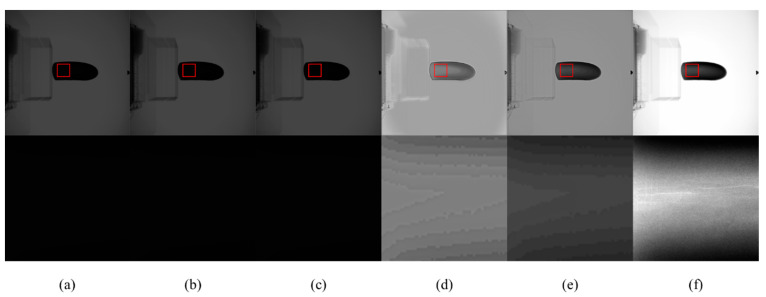
The contrast enhancement results of different methods. Top: raw image and the enhancement results; bottom: zoomed-in views on the highlighted area. (**a**) The raw images; (**b**) CLAHE; (**c**) LCM-CLAHE; (**d**) Retinex-Net; (**e**) Zero-DCE; and (**f**) our method.

**Figure 16 sensors-23-02498-f016:**
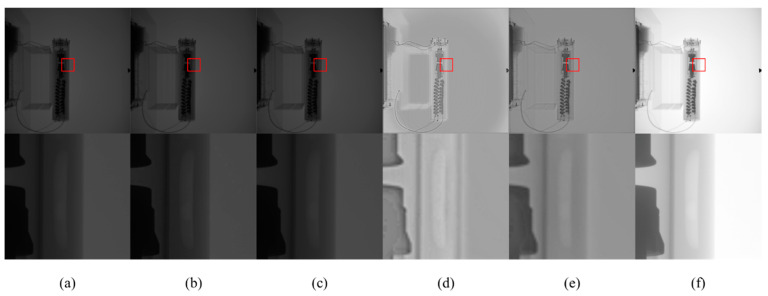
The contrast enhancement results of different methods. Top: raw image and the enhancement results; bottom: zoomed-in views on the highlighted area. (**a**) The raw images; (**b**) CLAHE; (**c**) LCM-CLAHE; (**d**) Retinex-Net; (**e**) Zero-DCE; and (**f**) our method.

**Table 1 sensors-23-02498-t001:** Image contrast of Fe (20 mm thick) and Al (1 mm thick).

	Filter Thickness (mm)	L_min_	L_max_	Contrast Ratio
150 kV	0	449	---	---
	0.5	219	44,475	203
	0.75	180	34,598	192
	1	161	28,064	174
180 kV	0	1135	---	---
	0.5	523	---	---
	0.75	425	---	---
	1	345	41,229	120

**Table 2 sensors-23-02498-t002:** Evaluation indicators of different methods.

		Original Image	CLAHE	LCM_CLAHE	RetinexNet	Zero_DCE	Proposed Method
90 kV	NRSS	0.7791	0.8599	0.7791	0.7615	0.7969	**0.8668**
	AG	0.614	0.814	0.614	0.215	0.256	**0.815**
	SF	4.7	4.72	4.7	4.39	4.62	**5.85**
	STD	51.22	50.47	51.22	27.17	44.4	**73.17**
	Entropy	6.5549	6.5677	6.5549	5.9134	6.8349	**7.013**
120 kV	NRSS	0.9457	0.9189	0.9457	0.9194	0.9457	**0.9838**
	AG	0.285	0.349	0.285	0.562	0.616	**0.623**
	SF	2.49	2.53	2.49	3.98	4.54	**5.4**
	STD	40.52	40.07	40.52	22.68	37.72	**87.19**
	Entropy	4.595	4.5985	4.595	4.6459	5.2104	**5.7642**
150 kV	NRSS	0.9663	0.9186	0.9663	0.9317	0.9341	**0.9881**
	AG	0.208	0.257	0.208	0.303	0.387	**0.566**
	SF	1.96	2.01	1.96	4.83	4.51	**5.26**
	STD	31.21	31.23	31.21	19.01	31.38	**82.33**
	Entropy	3.5394	3.5842	3.5394	3.9537	4.0467	**4.1662**
180 kV	NRSS	0.9428	0.9774	0.9428	0.8996	0.8086	**0.9856**
	AG	0.203	0.248	0.203	0.507	0.512	**0.578**
	SF	1.7	1.77	1.7	5.12	5.7	**6.03**
	STD	25.28	25.28	25.28	17.32	26.31	**75.05**
	Entropy	3.3953	3.5736	3.3953	3.9332	4.1628	**4.8621**

**Table 3 sensors-23-02498-t003:** Evaluation indicators of different methods.

		Original Image	CLAHE	LCM_CLAHE	RetinexNet	Zero_DCE	Proposed Method
Average value	NRSS	0.8882	0.9121	0.9073	0.9479	0.9365	**0.9723**
	AG	0.349	0.406	0.349	0.7813	0.7926	**0.8786**
	SF	1.86	1.926	1.86	3.9667	3.6866	**4.6766**
	STD	24.456	24.163	24.524	22.013	30.37	**63.636**
	Entropy	4.9966	5.001	4.9996	4.3561	5.3391	**6.2685**

## Data Availability

Not applicable.

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
