# Peer review of "X-ray Single Exposure Imaging and Image Processing of Objects with High Absorption Ratio"

_sensors, 2023, doi:10.3390/s23052498_

Round 1

Reviewer 1 Report

This paper discusses a proposed method based on Retinex theory to enhance imaging of X-ray single exposure images of high absorption ratio objects. It is a very interesting nowadays topic with encouraging results. It deserves to be published. However, I have several questions for the authors:

1.       Could you describe the setup for X-ray imaging? What kind of X-ray tube was used? What is the imaging device? What is the geometry and position of the X-ray tube, imaging sample and device?

2.       Is the X-ray energy spectrum for an operation voltage of 180 kV shown in Fig.1 measured or calculated?

3.       How did you calculate the X-ray attenuation length for Fig. 1? How do you compare the data with NIST Standard Database? https://www.nist.gov/pml/x-ray-mass-attenuation-coefficients

4.       As shown in the legend in Fig. 2, the step difference for Al is 2mm with No. of steps of 10. However, the photo for Fig. 2 (a) show only 8 steps.

5.       Why only 9 steps were observed in Fig. 3(a) and (c) for the iron imaging, while there are 10 steps of irons shown in Fig. 2(a)?

6.       On Page 4 Line 142, “When the thickness of the filter reaches 0.4mm” However, in Fig. 3 the filter thickness ranges from 0.5 to 1.0 mm. In the same sentence, “achieving a single exposure imaging of 20mm iron and 1mm aluminum” Is the minimum thickness of Al 1mm or 2mm as suggested in Fig. 2?

7.       For Fig. 5, could you specify what thickness of iron and aluminum is used?

8.       Could you explain why the contrast ratio for 90 kV is the highest without filter while the lowest with 1mm filter?

Reviewer 2 Report

The article is on a very good level.
There are only some minor observations.
The chapter name X-ray spectral bandwidth compression and imaging should be on the next page.
I recommend using black colour for the axis descriptions, not a grey one.
The description of Figure 8 should be on the same page as the figure itself.
The chapter name Qualitative analysis of X-ray images of objects with different absorption ratios should be on the next page as well.
Indenting the text after chart number 2 by one line would be more suitable.
Check the chapter name Qualitative analysis on lines 328, 340, and 360.
Check the spacing between text and semicolons regarding figures.
For instance, there is a missing space after "(e)" in the Figure 13 description.
When it comes to certain figures, a space before and after would result in a better arrangement.
Consider increasing the size of curves in some of the figures, for better legibility, for example, number 3 and number 5.

Perform an English check, look for possible missing articles in particular.
